# Extracellular Vesicles as Drug Delivery Systems in Organ Transplantation: The Next Frontier

**DOI:** 10.3390/pharmaceutics15030891

**Published:** 2023-03-09

**Authors:** Harry V. M. Spiers, Lukas K. J. Stadler, Hugo Smith, Vasilis Kosmoliaptsis

**Affiliations:** 1Department of Surgery, University of Cambridge, Addenbrooke’s Hospital, Cambridge CB2 0QQ, UK; hs781@cam.ac.uk (H.V.M.S.);; 2NIHR Blood and Transplant Research Unit in Organ Donation and Transplantation, University of Cambridge, Cambridge CB2 0QQ, UK

**Keywords:** extracellular vesicles, exosomes, drug delivery systems, ischemia reperfusion injury, allograft rejection, organ transplantation, machine perfusion

## Abstract

Extracellular vesicles are lipid bilayer-delimited nanoparticles excreted into the extracellular space by all cells. They carry a cargo rich in proteins, lipids and DNA, as well as a full complement of RNA species, which they deliver to recipient cells to induce downstream signalling, and they play a key role in many physiological and pathological processes. There is evidence that native and hybrid EVs may be used as effective drug delivery systems, with their intrinsic ability to protect and deliver a functional cargo by utilising endogenous cellular mechanisms making them attractive as therapeutics. Organ transplantation is the gold standard for treatment for suitable patients with end-stage organ failure. However, significant challenges still remain in organ transplantation; prevention of graft rejection requires heavy immunosuppression and the lack of donor organs results in a failure to meet demand, as manifested by growing waiting lists. Pre-clinical studies have demonstrated the ability of EVs to prevent rejection in transplantation and mitigate ischemia reperfusion injury in several disease models. The findings of this work have made clinical translation of EVs possible, with several clinical trials actively recruiting patients. However, there is much to be uncovered, and it is essential to understand the mechanisms behind the therapeutic benefits of EVs. Machine perfusion of isolated organs provides an unparalleled platform for the investigation of EV biology and the testing of the pharmacokinetic and pharmacodynamic properties of EVs. This review classifies EVs and their biogenesis routes, and discusses the isolation and characterisation methods adopted by the international EV research community, before delving into what is known about EVs as drug delivery systems and why organ transplantation represents an ideal platform for their development as drug delivery systems.

## 1. Introduction

Extracellular vesicles (EVs) are a heterogeneous group of membrane-derived vesicles released into the extracellular space by cells [1,2]. They play important roles in cell-to-cell communication, functioning as important intermediaries of cell signalling due to their unique biosignatures [3,4,5], and act as key mediators of many physiological processes [6]. EVs are released from all cell types studied to date, including endothelial cells [7,8], epithelial cells [9] and mesenchymal stem cells [10,11]; hence, they can also be found in a number of biofluids, including blood [12], synovial fluid [13], bile [14], breast milk [15], cerebrospinal fluid [16] and urine [17]. EVs have also been implicated in a number of diseases, including cardiovascular [18,19,20] and liver diseases [21,22], autoimmune [23] and neurodegenerative conditions [24,25,26] and cancer [27,28,29,30].

EVs are delimited by a lipid bilayer decorated with markers of various biogenesis routes prior to their secretion from the host cell [1]. They measure between 40 nm and 2000 nm, and their further subdivision by size is used to classify the main types of EVs: exosomes, microvesicles and apoptotic bodies (Table 1). Given the heterogeneous sizes and surface marker expressions of these nanoparticles, consensus on specific EV subtype markers remains elusive; therefore, assigning an EV to a specific biogenesis pathway is challenging. The International Society of Extracellular Vesicles (ISEV) has produced guidelines to enable EV researchers to determine the EV population at hand, along with its characteristics and purity [31]. Additional guidance published in the Minimal Information for Studies of EVs 2018 (MISEX2018) provides terminology that can be used to further describe EVs according to their physical characteristics, including size (e.g., small EVs (sEVs) <100 nm or <200 nm and medium/large EVs (m/lEVs) > 200 nm), density, the presence of specific markers (e.g., CD9+ EVs) and the condition (e.g., hypoxic EVs) or cell of origin (e.g., cholangiocyte-derived EVs). [31]

As the field of targeted drug delivery has expanded, the EV research field has begun to realise the potential of EVs as next-generation drug delivery vehicles. This review provides an overview of the subtypes and biogenesis routes of EVs, as well as methods of EV isolation and characterisation, before delving into the emerging roles of native and bioengineered EVs as drug delivery vehicles and, finally, addressing current real-world applications and challenges, with an emphasis on the field of organ transplantation.

## 2. Classification and Biogenesis of Extracellular Vesicles

Small EVs—those in the exosomal range of 40–150 nm—represent the smallest EV subtype and are derived by the endosomal pathway (Figure 1). They are generated by invagination of the late endosomal membrane, which is mediated by the endosomal sorting complex required for transport (ESCRT) proteins, including ESCRT-I, known as tumour susceptibility gene 101 (TSG101) [32]. Late endosomal membrane invagination forms intraluminal vesicles (ILVs) in a process orchestrated by ALG-2 interacting protein X (ALIX) after its direct interaction with syntenin-1, a multifunctional adapter protein that is essential for vesicle biogenesis, resulting in a specialised endosomal compartment known as a multivesicular body (MVB) [33]. MVBs fuse with the plasma membrane and exocytose the ILVs as exosomes. Tetraspanins (CD9, CD63, CD81) play a role in directing EV cargo during this process because of their involvement in recycling routes between cellular organelles and the plasma membrane [34,35]. As a result of this complex process, sEVs are enriched in the proteins that mediate their formation, including the ubiquitously expressed syntenin-1, as well as TSG101, ALIX and the variably expressed tetraspanins [1].

Large EVs (lEVs)—or microvesicles—have sizes in the range of 150 nm to 1000 nm, averaging at 250–400 nm [36,37]. Propelled by actomyosin-driven fission into the extracellular space, lEVs result from direct budding from the plasma membrane [38] (Figure 1). lEVs originate from sites of high membrane blebbing [39,40] where a host of factors modulate the deformability and bending of the membrane, including the organisation of the cytoskeleton, lipid composition and membrane fluidity alterations [38]. As a result of this, lEVs have unique lipid characteristics that result in increased uptake by recipient cells [41], such as externalisation of phosphatidylserine phospholipids [42]. Once loaded with cargo, the cone-shaped lipid ceramide promotes membrane bending [43] with the bleb “pinched” free from the cell through acto-myosin contraction [44], a tightly regulated process governed by the Rho family of GTPases [45].

Apoptotic bodies represent the largest EV subpopulation, measuring 1–5 μm. They form during programmed cell death upon disassembly of apoptotic membrane blebs (microtubule spikes [46], apoptodia [47] and beaded apoptodia [48]). These blebs contain carefully packaged cell contents, including organelles and DNA, which are phagocytosed by macrophages after release into the extracellular space [49]. As these vesicles contain numerous cellular components, they express markers that overlap with sEV and lEV populations; therefore, size is a key characteristic for their determination in samples [31].

The biogenesis of EVs produces an intraluminal cargo from the parent cell that is protected by their phospholipid membrane, allowing transfer of proteins, lipids, DNA and a full repertoire of RNA species (including mRNA, microRNA (miRNA) and other non-coding RNAs) [1,2]. The ability of EVs to protect a wide variety of nucleic acid species [2], carry these across biological barriers [50,51,52], utilise endogenous intracellular trafficking mechanisms and trigger a response in recipient cells [53] makes them attractive drug delivery systems for both endogenous and bioengineered compounds.

## 3. Isolation and Characterisation of Extracellular Vesicles

EVs must be isolated from their respective source samples and other vesicles, after which their size, shape and density can be characterised (Figure 2) [54]. The isolation of EVs from multiple cell sources is well-established, as demonstrated by the transcriptomic atlas of EV RNA from 20 biofluids [55]. However, isolation of specific subspecies remains challenging, given the heterogeneity of EVs in terms of size, density and cargo. Several approaches have been established to isolate EVs, with each requiring a trade-off between purity, EV recovery and the integrity of the isolated vesicle; commonly used methods for EV isolation and purification are outlined in Table 2. As there is no single optimal method of isolation, the choice of technique(s) is based on the downstream application. The ISEV recognises the challenges of EV purification and has issued the following recommendations: EV quantification should be undertaken with at least two methods, including protein amount and particle number. Assessment of transmembrane or glycosylphosphatidylinositol-anchored proteins and membrane-associated cytosolic proteins should then be performed. Contaminants should be evaluated, and images of single EVs obtained using wide-field and close-up assessments should be published [31].

Once isolated, EVs must be characterised to further identify the population present. EV characterisation is a multifaceted process comprising numerous established and emerging techniques, each with their own advantages and disadvantages. Recommendations from the ISEV set out in the MISEV2018 guidelines [31] suggest that contemporary EV studies quantify the sources of EVs in terms of the number of secreting cells, volume of biofluid and mass of tissue (where each is applicable). EV concentration should be assessed according to the total particle number. The typical EV content in the sample should be quantified (commonly by identifying transmembrane or cytosolic proteins) and non-vesicular contaminants tested for following isolation [31]. The size and concentration of EVs in the source sample are assessed using nanoparticle tracking analysis (NTA) and tuneable resistive pulse sensing (TRPS) [56]. NTA involves shining a laser into the sample as it passes under a video capture system and uses dynamic light scattering to calculate size based on the Brownian motion of a particle while calculating concentration by counting the number of particles per frame. TRPS involves driving EVs through a nanopore in an electrolyte fluid cell that is sampled 50,000 times per second using a combination of pressure and voltage, and each particle results in a resistive pulse or “blockage” signal that is detected and measured. The blockade magnitude is directly proportional to the volume of each particle, with the frequency of blockade being directly proportional to the sample concentration. It is also desirable to determine EV morphology, and this can be achieved with microscopy techniques, including transmission electron microscopy (TEM) [57] and cryo-electron microscopy [58]. Techniques employed for EV content include Western blotting for specific EV markers (including the aforementioned syntenin-1, TSG101 and ALIX [1]), flow cytometry [59,60], ExoView [61] and the enzyme-linked immunosorbent assay (ELISA) [62]. Additional high-definition analyses of EV content can be performed using techniques such as mass spectrometry proteomic analysis [63,64] and genomic sequencing for transcriptomic profiling [65].

Reliable and time-efficient EV isolation and characterisation protocols are required for consistent isolation of EVs with therapeutic properties and those that have been bioengineered to function as therapeutic agents. This is not currently the standard, and several issues require addressing to refine the process. Methods to achieve large-volume, high-purity EV isolation that are not expensive, complex or time-consuming and focus on discrimination between specific EV populations during isolation using their distinctive physicochemical or biological properties in order to generate preparations capable of delivering consistent therapeutic effects are essential.

## 4. Extracellular Vesicles as Therapeutics

EVs can be harnessed therapeutically and the pharmacological classification of EV-based therapeutics depends on the active substance [66]. They can be divided into three main categories: (1) natural EVs, which are native and obtained from genetically engineered cells; (2) hybrid EVs, which are post-modified with drugs or surface ligands; (3) EV-inspired liposomes. In this section, we focus on natural and hybrid EVs, as these are grouped together from a regulatory perspective [66].

### 4.1. Native EVs as Drug Delivery Systems

The idea of using native EVs for disease therapy, as a sort of “naturally occurring” therapeutic nanoparticle, arose originally from stem cell studies. Several groups investigating the potential therapeutic use of mesenchymal stem cells (MSCs) in the treatment of cardiovascular diseases found that the therapeutic effect of the intervention was not derived from the engrafted cells themselves but rather from their paracrine effectors; in particular, EVs [67,68,69]. Since those early studies, the use of stem and progenitor cell-derived EVs for the treatment of various diseases has been explored extensively.

Stem cell-derived EVs demonstrate numerous cardioprotective effects, which have made them prime candidates in the treatment of myocardial infarction (MI) and other ischemic events. Studies with animals have shown that EVs promote the survival of heart endothelial cells and cardiomyocytes following ischemic insults through the activation of common survival pathways, such as AKT and ERK [70]. Of particular interest is one recent study that demonstrated the superior effectiveness of the cardiac repair and safety associated with induced pluripotent stem cell (iPSC)-derived EVs compared to the iPSCs themselves [71]. In other studies, EVs have been shown to supress inflammation, decrease oxidative stress and even promote the formation of new blood vessels in animal models of MI [72,73,74]. EVs that demonstrate cardiovascular effects have been isolated from numerous cellular sources, including the aforementioned MSCs and iPSCs, as well as putative cardiac progenitor cells and even differentiated somatic cells [75,76,77,78].

The molecular candidates responsible for the mediation of these EVs’ therapeutic effects are most commonly found amongst the miRNA cargo of EVs, which represents the “drug” in this system. For example, extracellular matrix-derived EVs can regulate the acetylation of transcription factor GATA4 via miRNA-199a-3p, thus rescuing the electrical function in bioengineered atria [79]. Similarly, EVs isolated from the cardiosphere were able to ameliorate heart dysfunction in a mouse model of MI by decreasing inflammation and apoptosis via miRNA-146, as well as increasing cardiomyocyte proliferation and promoting angiogenesis [80]. Numerous other miRNAs have been shown to have cardiovascular protective effects in vitro and in animal models, including miRNA-24, miRNA-126, miRNA-181b and miRNA-294-3p [81,82,83].

Additionally, there are several EV cargo proteins for which roles in mediating therapeutic cardiovascular-protective effects have been demonstrated. Pappalysin-1 is expressed on the surface of EVs from cardiac progenitor cells and has been shown to mediate the pro-survival properties of those EVs through the activation of the IGF-1 receptor and subsequent phosphorylation of its downstream anti-apoptotic effectors AKT and ERK1/2 [84]. Another protein component of EVs with potential therapeutic activity is platelet-derived growth factor D, which has been shown to mediate cardiac regeneration and stimulate angiogenesis in a rat model of acute myocardial infarction [85].

While the therapeutic use of native EVs has so far been most extensively explored in cardiovascular research, there is growing interest in the areas of inflammatory and autoimmune diseases. In the field of dermatology, EVs are starting to garner attention as therapeutics for inflammatory skin diseases; a number of studies have shown that EVs can aid in the process of wound healing, particularly via the molecular actions of miR-21. EVs derived from keratinocytes were shown to promote wound healing by regulating fibroblast function and through stimulation of angiogenesis [86,87]. Similarly, in a recent study of umbilical cord-derived EVs, Liu et al. showed that wound healing could be accelerated by enhancing the proliferation and migration of fibroblasts. [88].

Another area of interest is represented by inflammatory diseases of the central nervous system, an area of the body that is particularly challenging to treat with stem cells due to restricted access via the blood–brain barrier (BBB). EVs, however, have been shown to readily cross the BBB in several studies [89]. In a model of preterm brain injury, EVs from MSCs slowed the signs of inflammation and improved signs of inflammation- induced neuronal degeneration [90]. In a separate study, Clark et al. showed that MSC-derived EVs from the placenta were able to slow disease progression of multiple sclerosis in an animal model of the disease [91].

In light of the proven anti-inflammatory properties of EVs derived from stem cells, their use in the treatment of autoimmune diseases is beginning to be investigated. In one recent example, Cosenza and colleagues isolated EVs from MSCs via ultracentrifugation and first tested their immunosuppressive effects in an in vitro lymphocyte-based assay [92]. They reported reduced T-cell proliferation upon EV exposure, as well as inhibition of plasmablast differentiation of B-cells. Furthermore, in a mouse model of collagen- induced arthritis, the authors found significantly decreased signs of inflammation [92]. Along similar lines, there is significant interest in finding more effective ways of alleviating the systemic symptoms of inflammation in patients with lupus erythematosus. To this end, different research groups have demonstrated the immunosuppressive properties of MSC-derived EVs in models of acute tubular inflammation in the kidney [93], as well as the efficacy of these nanoparticles in promoting cartilage repair in a mouse model of osteochondral defect [94].

Native EVs derived from stem cells and other cell sources have demonstrated clear applicability in the treatment of various pathologies, particularly those with a significant inflammatory component. Advantages over other treatment methods (such as whole cells) include their low immunogenicity (vs. whole cells), their small size (vs. whole cells) and the ease of large-scale production (vs. engineered biomolecules). There are numerous human diseases, however, where a more targeted delivery of the active agent is desirable in order to minimise off-target effects or to deliver a more potent dose with precision.

### 4.2. Bioengineered Extracellular Vesicles as Therapeutics

The therapeutic efficacy of EVs can be improved synthetically through various strategies; for example, by adding a targeting molecule (antibody, aptamer, etc.) to the EV surface, by loading the nanoparticle with a specific biologically active agent or by extending the time EVs spend in circulation through chemical modifications, creating a so-called “hybrid EV” [95].

One obstacle standing in the way of widespread utilization of EVs in disease therapy is the challenge of ensuring accurate delivery to the site of action while preventing off-target effects. In order to equip EVs with specific targeting molecules, two different strategies have been explored: genetic modification of the producing cell (e.g., specific cell surface receptor expression) and direct chemical modification of the purified EV. One example of the first strategy is the genetic modification of EV-producing cells to express recombinant EGFR-specific nanobodies with a glycosylphosphatidylinositol (GPI)- anchoring peptide. As EVs carry high levels of GPI, the nanobodies were enriched on the EV surface, which, in turn, equipped the EVs with targeting specificity for EGFR cells [96]. While this method resulted in successful targeting of the EVs toward their intended sites of action, the genetic modification of EV-producing cells may not always be a viable option in future EV therapy due to the time-consuming procedure and the challenges of genetic modification of cells derived from a patient’s body fluids.

The strategy of chemically modifying EVs post-purification makes use of the EV membrane composition to specifically attach a targeting moiety with high density and in a controllable manner. One such approach made use of the high levels of phosphatidylserine (PS) in the lipid bilayer of EVs, genetically fusing a targeting peptide to the C1C2 domain of lactadherin, a high-affinity binder of PS [97,98]. Specifically, the group attached an anti-EGFR nanobody to the cell surface of EVs in this manner and found that it resulted in dose-dependent uptake of EVs by EGFR^+^ cells. Crucially, addition of the fusion peptide did not compromise the integrity or size of the EV, thus further supporting the method’s suitability for potential therapeutic uses. Another group made use of the popular click-chemistry—in this case, copper-free azide alkyne cyclo-addition—to furnish MSC-derived EVs with a high affinity integrin α_v_β_3_-binding peptide [99]. These bioengineered EVs were able to target the ischaemic region of the brain in a mouse model of artery occlusion. Furthermore, the authors reported reduced inflammatory response in these animals versus control animals treated with non-specific EVs.

As well as their use as mediators of cell targeting, various peptides have also been used to promote internalisation of EVs into target cells. Nakase and co-workers made use of a sulfo-N-ε-maleimidocaproyl-oxysulfosuccinimide ester chemistry to decorate the surface of EVs with an arginine-rich micropinocytosis-inducing peptide. This modification resulted in substantially increased uptake into CHO cells and improved delivery of the EVs’ cytotoxic saporin protein cargo [100].

Another area of scientific interest is the modification of the bioactivity of EVs (most commonly by customising the cargo they carry), which would make it possible to adapt their uses to different diseases. One prominent example is the use of EVs as drug delivery vehicles in cancer therapy, whereby engineered EVs would be specifically targeted toward a tumour and deliver a cytotoxic agent. As with the different targeting methods discussed above, loading an EV with a drug or bioactive molecule could be achieved either prior to EV production (i.e., through modification of the cell of origin) or after the EVs have been isolated. Opting for the first approach, Tang at al. incubated their cell line of choice with four different chemotherapeutic agents (including cis-platin and methotrexate) by simply adding those molecules to the cell culture medium for 12 h. Following isolation of the EVs from the cell culture medium via ultracentrifugation, the authors went on to show that they had cytotoxic actions affecting the target cancer cell line [101].

Conversely, EVs can be isolated first then loaded with drugs. Different techniques have been investigated depending on the molecule of choice. In the case of hydrophobic entities or small-molecule drugs, passive diffusion through the EV surrounding lipid bilayer may be possible. Wei et al. isolated EVs from bone marrow MSCs and loaded them with doxorubicin through overnight incubation. Following a dialysis step to ensure removal of free drug, the authors verified the integrity of these modified EVs using NTA and TEM and then tested their activity in a cell proliferation assay. Additionally, they assessed EV cellular uptake using flow cytometry. The authors found high levels of the modified EVs in the target cells, as well as significant growth inhibition [102].

Other methods used to load EVs with anti-cancer agents include sonication, electroporation and treatment with cellular transfection reagents, all of which work by disrupting the vesicle membrane and allow uptake of drug molecules [103]. Examples of their use come from the field of cardiovascular diseases. One group used a custom-made EV transfection reagent—ExoFect—to enrich human peripheral blood-derived EVs with an miRNA-21 inhibitor. This miRNA plays a crucial role in the development of fibrosis after an MI, and use of those modified EVs in a mouse model of MI resulted in significantly reduced fibrosis [104]. In a separate study, Youn et al. employed electroporation to load cardiac progenitor cell-derived EVs with miRNA-322, followed by systemic injection into a mouse model of MI. The intervention resulted in increased angiogenesis and improved recovery in the animals [105].

In sum, EVs clearly have strong potential for widespread use in disease therapy, whether is in their “native” form, exerting anti-inflammatory effects, or as drug delivery vehicles. One possible drawback, however, is the fact that unmodified EVs are subject to rapid circulatory clearance and demonstrate relatively low accumulation in target tissues [106]. The latter issue can be overcome through addition of targeting moieties as described above. In order to address the issue of rapid clearance, EVs can be coated with polyethylene glycol (PEG). PEG is a hydrophilic polymer and has previously been shown to increase the circulation time of nanoparticles [107]. With regard to EVs, it has been reported that PEGylation increases EV circulation time, as well as reducing non-specific interactions with cells [108].

While native, unmodified EVs have shown broad applicability in the pre-clinical treatment of inflammatory diseases, the potential uses of these nanoparticles in the treatment of disease can be significantly expanded through chemical and synthetic modification.

## 5. Extracellular Vesicles in Organ Transplantation

Given the evidence that native and hybrid EVs are promising drug delivery systems, it is no surprise that these nanoparticles have made their way into clinical practice, being employed for real-world uses as therapeutics in several on-going clinical trials (Table 3). Organ transplantation is the gold standard for treatment for suitable patients with end-stage organ failure. Kidney transplantation, for example, has become a routine procedure because it confers improved length and quality of life in patients, as well as having clear economic benefits [109]. Nevertheless, significant challenges still remain in organ transplantation, including graft rejection requiring extensive immunosuppression and the lack of donor organs available to meet the ever-growing waiting lists [110]. As discussed, the inherent ability of EVs to facilitate the transfer of protein, lipid and RNA payloads to target cells, crossing natural barriers and utilising intrinsic cellular machinery and pathways, makes them an attractive therapeutic vehicle. Additionally, EVs are potentially less immunogenic due to the reduced surface expression of human leucocyte antigens compared to cellular therapies. They are also unable to replicate, which reduces the risk of tumour generation subsequent to their delivery, and their inherent targeting mechanisms reduce off-target effects [53]. These factors make them a promising and realistic clinical drug delivery system for development in organ transplantation, a field associated with unmet needs in terms of donor organ utilisation and long-term graft function whilst also representing a unique platform for the study of EVs and their utilisation as therapeutics.

### 5.1. Alloimmune Response Modulation

There is increasing evidence that priming and effector mechanisms behind graft rejection are mediated by EVs. Dendritic cells (DCs), important antigen-presenting cells, may be immature or mature and EVs from either population can mediate the opposite effects. In a mouse skin graft model, it appears that immature DC-derived EVs promoted T cell activation without rejection, whilst EVs from mature DCs contrastingly induced effector T cells that led to skin graft rejection [111]. The same group from France characterised immature DC-EVs, which express low levels of ICAM-1, MHC-II, CD89 and CD86 and may mediate suppressive functions [111]. Li et al. went on to demonstrate that EVs from donor-derived immature DCs prolonged graft survival in a mouse model of cardiac transplantation [112]. The proposed effector mechanism involved IL-10 induction, favouring FoxP3 expression in the T cell compartment in addition to reducing interferon-gamma (IFN-y) and IL-17 mRNA. Further studies have corroborated these findings, demonstrating the ability of immature DC derived-EVs to promote short-term graft survival, with the best results obtained when EVs were given alongside immunosuppression or T regulatory cells [112,113,114,115]. B cells have been known to secrete EVs [116]; however, it has been shown that they do not secrete them constitutively but rather upon B call activation via appropriate cell signalling [117]. Stimulation of B cells via CD40 and IL-4 results in secretion of an sEV population expressing MHC classes I and II, as well as B cell receptor components and tetraspanins [117]. Relevant to transplantation, sEVs released by activated B cells can be acquired by DCs, which subsequently prime cytotoxic T lymphocytes (CTLs) with the help of CD4+ T cells and natural killer cells; CTLs are known to play an important role in T cell-mediated graft rejection [118].

Whist there is evidence of the mechanisms whereby EVs mediate graft rejection, they can also achieve an immunosuppressive effect and promote immune tolerance, a key therapeutic target in transplantation. In a rat model of kidney transplantation, T regulatory (Treg) derived-EVs inhibited T cell proliferation [119]. The EVs secreted by mouse Tregs were shown to have a specific miRNA profile that led to downstream immunosuppressive effects in other immune compartments; miR-150-5p and miR-142-3p were associated with reduced IL-6 secretion by DCs and increased secretion of anti- inflammatory IL-10 whilst suppressing IFN-y secretion by CD4^+^ T cells [120,121]. The ability of Treg-derived EVs to reduce proliferation of CD4^+^ T cells and their subsequent secretion of IL-2 and IFN-y has been demonstrated to result from the presence of the ectoenzyme NT5E or CD73, which acts via binding of the adenosine receptor A2aR to facilitate the immunosuppressive effects of Tregs [122,123,124,125]. This may translate into prolonged graft survival and effector T-cell proliferation inhibition, as suggested in a renal transplant model of acute rejection. Treg-derived EVs also suppress the proliferation and viability of rejection-mediating CTLs, reducing their ability to produce perforin and IFN-y [126]. This has been applied to a rat model of liver transplantation, where Treg EV-treated rats demonstrated improved short-term survival compared to their non-treated counterparts [126]. These studies highlight the potential therapeutic role of EVs derived from both innate and humoral immune compartments; in this context, EVs may have a role as adjuncts to immunosuppressive therapies. In addition, specific miRNA species, similar to those described above, could be packaged into EVs for delivery as further adjuncts to immunosuppression regimes or even to treat graft rejection. Further work is clearly needed to transfer these promising findings from animal models to clinical practice.

### 5.2. Ischemia-Reperfusion Injury

A major challenge in organ transplantation is ischemia-reperfusion injury (IRI); a tissue injury that occurs when the blood supply to organs is interrupted and then restored [127]. Waiting lists for organ transplantation are growing and the availability of donor organs does not meet the demand [110], leading to an increase in the utilisation of suboptimal (extended criteria or marginal) organs in an attempt to meet needs and reduce waitlist mortality [128,129]. However, marginal organs are more susceptible to ischemia-reperfusion injury (IRI) and an enhanced pathological response to the restoration of oxygenated blood supply, thereby leading to suboptimal graft function [130,131,132], as well as increased graft immunogenicity [133] and, subsequently, increased rates of allograft rejection [134].

IRI prompts the release of inflammatory mediators and reactive oxygen species, leading to tissue damage, necrosis and release of damage-associated molecular patterns (DAMPs) [135]. DAMP recognition by innate immune cells, such as macrophages, activates the nuclear factor-kappa beta (NF-kB) pathway responsible for transcriptional induction of pro-inflammatory mediators [136]. These include inflammasomes, which control the post-translational proteolytic activation of pro-inflammatory cytokines to orchestrate pyroptosis, a pro-inflammatory form of cell death, which is the ultimate outcome of IRI [136]. NF-kB signalling has been correlated with the release of EVs [137]. Pre-clinical studies have shown that EVs released during IRI are central to immune activation and propagation of inflammation, commonly through interactions with the NF-κB signalling pathway; for example, through upregulation of IL-1B, a potent pro- inflammatory cytokine [138]. There is evidence at the transcriptomic level that human kidneys suffering prolonged delayed graft function post-kidney transplant demonstrate upregulation of NF-kB-mediated signalling pathways [139]. This has been reproduced in a rat model of liver transplantation [140], where mRNA levels of NF-kB pathway mediators, including TNF-alpha, IL-1B and IL-10, were significantly raised during IRI [140], highlighting this inflammatory transcription regulator as a key player in organ IRI.

EVs are a promising therapy for IRI and increasingly studied in pre-clinical models. Human induced pluripotent stem cell (hiPSC)-derived MSC-EVs ameliorated rat kidney IRI through the delivery of specificity protein 1 into target renal cells [141]. More specific mechanisms of IRI have also been targeted by EVs utilising EXPLOR, a novel optogenetically engineered exosome technology. In this study, Kim et al. engineered EVs to deliver super-repressor inhibitor of NF-kB (ExosrIkB), demonstrating significantly improved outcomes in a mouse kidney IRI model [142]. The treated group achieved lower blood levels of urea, creatinine and neutrophil gelatinase-associated lipocalin (a biomarker for renal injury released by renal tubular cells under stress) compared to the naive control group. This was confirmed at the transcriptomic level with reduced gene expression of pro-inflammatory cytokines and adhesion molecule in the treated group [142]. Administration of human cord blood endothelial colony-forming cell (ECFC)- derived EVs protected mice against kidney IRI; exosomes were found to be enriched for miR-86-5p, which inhibits the phosphatase and tensin homolog (PTEN) and enhances Akt phosphorylation [143]. Through these mechanisms, miR-86-5p inhibited hypoxia-induced apoptotic responses, protecting against IRI induced renal injury. Through an alternative mechanism, miR-199a-3p carried by MSC-derived EVs also inhibited apoptosis in a kidney IRI model, this time through downregulation of semaphoring 3A in addition to Akt and ERK pathway activation [144]. Endoplasmic reticulum stress during hypoxia is a known consequence of IRI, leading to misfolding of proteins and cellular dysfunction. miR-199a-5p transfer in bone marrow stem cell-derived sEVs targeted binding immunoglobulin protein to reduce ER stress and restore cellular homeostasis during early reperfusion, mitigating IRI [145].

In the liver, Du et al. demonstrated that MSCs derived from hiPSCs conferred protection against mouse hepatic IRI via activation of sphingosine kinase and the sphingosine-1-phosphage signalling pathway, as well as promoting cell proliferation by the same pathway [146]. The treated group achieved lower alanine and aspartate aminotransferase levels, as well as histological reductions in hepatocyte necrosis and congestion, with increased expression of proliferation markers. In another pre-clinical study, bone marrow-derived DC EVs upregulated anti-inflammatory cytokines TGF-beta, FoxP3 and IL-10, alleviating liver IRI in mice [147]. Activation of Toll-like receptor signalling by DAMPS generated during IRI can induce formation of neutrophil extracellular traps (NETs), the formation of which has been verified as a critical step in liver IRI [148]. Human umbilical cord-derived MSC-EVs are able to transfer functional mitochondria to intrahepatic neutrophils, subsequently maintaining mitochondrial quality through mitochondrial fusion and leading to inhibition of NET formation in a cell culture model of IRI [149]. Steatotic livers are particularly susceptible to IRI and suffer from ferroptosis, a process of iron-dependent cell death [150]. Haem-oxygenase-1 (HO-1) is an antioxidant that can protect cells from oxidative stress; delivery of EVs from bone marrow-derived MSCs overexpressing HO-1 suppressed ferroptosis in a rat model of liver transplant IRI via EV-mediated transfer of miR-29a-3p [151]. miR-29a-3p targeted the iron response element-binding protein 2 (Ireb2), the main regulator of cellular iron homeostasis, leading to a reduction in cellular iron and downregulation of the transferring receptor, preventing further iron uptake by cells [151].

Myocardial IRI has been studied extensively and mechanisms of EV-mediated IRI amelioration that may be translatable to the field of organ transplantation have been highlighted. Rat MSC-derived EVs abolished the production of reactive oxygen species (another mediator of IRI) and improved cardiac function in vitro [152]. They also saw reduced apoptotic activity through autophagy promotion via adenosine monophosphate-activated protein kinase (AMPK) and AKT pathways. In oxygen-glucose deprivation cell culture models, dental pulp-derived MSC-derived EVs containing miR-4732-3p prolonged spontaneous beating, lowered reactive oxygen species and reduced apoptosis in cardiomyocytes [153]. The benefits of miR-4732-3p persisted when the same cells were transplanted into infarcted rat hearts with reduced scar tissue and preserved cardiac function [153]. Villa Del Campo et al. utilised a novel model of human myocardial injury in the form of cryoinjured engineered human myocardium to test the ability of epicardial EVs to promote cardiomyocyte proliferation [154]. The group identified the EV cargos miR-30a, miR-100, miR-27a and miR-30e as the mediators of the observed effect. Although an IRI model was not utilised, these miRNAs demonstrated reductions in apoptosis via Akt and ERK pathway activation, as seen in previously discussed kidney models of IRI. EVs have also been shown to reduce inflammatory cytokines in lung IRI in mice. The reductions in inflammatory cytokines TNF-alpha, IL-1B, IL-6, IL-7 and IL-8 from intratracheal administration of EVs from MSC-EVs, which were coupled with a reduction in lung oedema and lower M1 macrophage polarisation of alveolar macrophages, were suggested to be mediated by transfer of miR-21-5p encapsulated within EVs [155]. Human Wharton jelly mesenchymal stem cell (hWJMSC)-derived EVs have been utilised in the mitigation of lung IRI in a murine lung transplant model with both in vivo and in vitro components [156]. The group were able to achieve significant attenuation of lung dysfunction and injury (reduced oedema, myeloperoxidase levels and neutrophil infiltration) in the group treated with MSC-derived EVs compared to the untreated control group. They further observed significant reductions in proinflammatory cytokines TNF-alpha, IL-17, CXCL1 and HMGB1 [156]. Delving further into the mechanism behind the injury and its treatment, they were able to show downregulation of primary invariant natural killer T cell-produced IL-17 and macrophage-produced TNF-alpha and HMGB1 after IRI cycles [156]. Whilst many studies have focused on mitigating IRI through EV-delivered miRNAs, Cai et al. delivered an miRNA inhibitor packaged into EVs to inhibit the effect of a specific miRNA implicated in IRI [157]. Having identified that miR-206 expression was increased in the bronchoalveolar lavage fluid of patients on days 0 and 1 after lung transplantation, they replicated this finding in a mouse model of IRI resulting from lung transplantation. Delivery of antagomir-206-enriched EVs attenuated lung dysfunction, injury and oedema compared to treatment with EVs alone after murine lung IRI. The enriched EV-treated group also showed decreased expression of pro- inflammatory cytokines and chemokine-ligant-1, a potent chemoattractant for several groups of immune cells that propagate tissue injury [157].

There is clear evidence that EVs (generally derived from stem cells) can be harnessed to attenuate IRI. It is essential that the mechanisms behind their action are understood to allow targeting of specific cellular processes. The specific miRNAs that modulate cellular processes and homeostasis are promising therapeutic targets for future exploration, as are EV-packaged antagonists of cellular miRNAs implicated in IRI. Other biologically active EV cargos mediating their IRI protective effects must be uncovered in order to design appropriate therapeutics. Furthermore, the timing of the delivery of EV-based therapeutics in the treatment of IRI must be explored, with evidence showing that both pre-ischemia and post-reperfusion interventions are possible. Studies are required in human organ systems to understand the mechanisms of IRI and thus enable expansion of therapeutic targeting, as well as to promote clinical translation of EV drug delivery systems.

### 5.3. Machine Perfusion as a Platform for EV Drug Delivery Systems

Machine perfusion of organs outside the body prior to transplantation has been introduced to mitigate IRI and not only enables graft viability assessment [158,159] but also the delivery of therapeutics for the repair of organs prior to transplantation (Figure 3) [160]. The efficacy of MSC delivery during perfusion to mitigate IRI has been demonstrated in several organ systems [161,162,163]. Building on the earlier discussion on native EVs as drug delivery systems, EVs from stem cells have also been delivered during organ perfusion with promising results. In a recent study [164], human liver stem cell-derived EVs were delivered with a rat liver model of IRI during normothermic machine perfusion, and attenuation of IRI was demonstrated via improved biochemical markers of liver injury and a histologically assessed reduction in necrosis. A second group from Italy demonstrated the ability of MSC-derived EVs delivered during hypothermic oxygenated perfusion of the kidneys from extended criteria donors to reduce IRI. The treated kidneys demonstrated improved global renal ischemia damage scores, lower apoptotic markers and lower lactate levels [165]. The machine perfusion model provides an ideal platform to test new therapies, such as EV drug delivery systems, with isolated organs. For example, the liver is an ideal organ for the examination of EV therapy, as the vast majority of systemically administered EVs accumulate in the liver [166,167]. EVs can be delivered into the perfusate, directly into the parenchyma itself or even through secretory apparatuses, such as the bile duct in the liver or the ureter in kidneys, providing a variety of delivery methods to implement in trials and optimise. Not only does isolated organ perfusion have the capacity to provide important mechanistic insights concerning EVs, but it is ideal for pharmacological testing [168] and directly translatable to EV drug delivery system assessment. Organ transplantation represents an important clinical field for therapeutic intervention, offering a clear platform enabling simple delivery, rigorous testing and expansion of the use of both native and hybrid EV drug delivery systems.

### 5.4. Barriers to Clinical Translation

EVs are an attractive drug delivery system because of their ability to reduce the toxic effects that can result from the introduction of a foreign substance into the body. However, larger scale and more specialised clinical translations of EVs remain challenging due to the inherent complexity of the nanoparticles themselves, the heterogeneity in their sizes and natural (batch-to-batch) variations during production, which result in higher intrinsic risks in the production process than for their purely synthetic liposomal counterparts. The production of EV-based drug delivery systems faces several challenges, including a degree of inherent biological variability potentially resulting in product heterogeneity. There are two main areas where this variability may be introduced: firstly, as part of the upstream processing (i.e., inside the cells used to express the desired biological effects) and, secondly, in the downstream processing (i.e., through the manufacturing process itself) [169]. For EVs produced by cells, the culture conditions of the cells strongly influence the product quality—not only in terms of yield but also in terms of composition and subsequent bioactivity [170]. Another important consideration is that EVs are a relatively static product and, except for degradation, they are not expected to change post-harvesting. Therefore, minor changes that could have considerable impact on product quality and activity are more challenging to identify. Ensuring good manufacturing practice is essential and, whilst known process controls from fields such as cell-based therapies could be adapted, the sizes and unique complexity of EVs mean that additional process controls are required. Despite these predicaments, there is significant enthusiasm toward the continuation of the clinical translation of EVs, which has led to a position paper from the ISEV regarding the use of EVs in clinical trials [66]. They build on the current practice followed by manufacturers of biological pharmaceutics, who routinely identify all substances in a drug that exert a particular metabolic, immunological or pharmacological action and are responsible for the drug’s biological (therapeutic) effects. If all the therapeutic effects of an EV intended for drug delivery were to be attributed to the loaded molecule and not the EV itself, then the EV would be considered a non-active component; accordingly, it would not be necessary to determine the mechanism of action but only the safety profile of the EV preparation.

Furthermore, the biologically active EVs that can be produced must be understood both in terms of biological cargo and the downstream biological processes they are linked to. Consistent isolation of biologically active sEVs—i.e., exosomes—is extremely challenging due to the lack of specific markers distinguishing them from other sEV species. A recent reassessment of exosome composition in an elegant study utilising high-resolution density gradient fractionation and direct immunoaffinity capture highlighted the presence of annexin A1 and A2, calcium-dependant phospholipid binding proteins, in non-exosomal small to large EVs [1]. Whilst the absence of larger vesicle markers, along with other characteristics, suggests the presence of exosomes, clinical translation requires certainty about the EV subpopulation being used. The ability to isolate single EVs has already been employed clinically, such as in Single Particle Automated Raman Trapping Analysis (SPARTA), which uses EVs as breast cancer biomarkers [171]; further methods of exploring the surface expression of sEVs are needed to identify exosome surface markers. Once identified, the next important step towards clinically translating exosomal subpopulations would be understanding their biological functions. Many studies have utilised a variety of protocols to purify EVs in the exosomal range, but none have allowed reliable association with a function or group of functions. Novel technology, such as phage-based display technology for the identification and isolation of disease-released exosomes, could be used to capture the biological implications of exosome production and release under different conditions [172]. Further progress requires the development of new technologies for the association of a specific marker with an exosome subtype and robust demonstration of a correlation with biological function.

In addition to the barriers outlined above, further research is required before EVs can be routinely used in transplantation. Whilst the mechanisms of graft rejection and IRI have now been explored in depth, there is still a lot to be uncovered and it is essential for the mechanisms behind the therapeutic benefits of EVs to be understood. Modulating the upstream pathways of inflammation, for example, has downstream effects on multiple effector mechanisms that are potentially important in other cellular processes and regulatory mechanisms. Mechanistic insights are essential to ensure the potential inhibition and activation of cellular pathways mediated by EV cargo do not have secondary negative effects unexplored in pre-clinical models thus far.

## 6. Conclusions and Future Perspectives

There is evidence that native and hybrid EVs may be used as effective drug delivery systems, with their intrinsic ability to protect and deliver a functional cargo by utilising endogenous cellular mechanisms making them attractive as therapeutics. Pre-clinical studies have demonstrated the potential of EVs in preventing rejection in transplantation and mitigating ischemia reperfusion injury in several models. This work has enabled clinical translation of specific (generally stem cell-derived) EV populations and led to several clinical trials actively recruiting patients, the results of which are eagerly awaited. However, EVs’ rapid and widespread clinical translation is challenged by the lack of cost-effective, large-scale isolation and characterisation methodologies. The emergence of high-sensitivity characterisation methods is expected to provide important insights into the biology of these powerful nanoparticles, with that improved understanding translating into increased utilisation in clinical practice. Organ transplantation represents an area that has the potential to significantly benefit from EV-based therapeutics. Machine perfusion of isolated organs provides an unparalleled platform for the investigation of EV biology and testing of their pharmacokinetics and pharmacodynamics. The global population of patients with end-stage organ failure, as well as those facing it, will benefit significantly if we can deepen our understanding of thus far unexplained EV-mediated mechanisms and harness EVs’ potential as drug delivery systems.

## Figures and Tables

**Figure 1 pharmaceutics-15-00891-f001:**
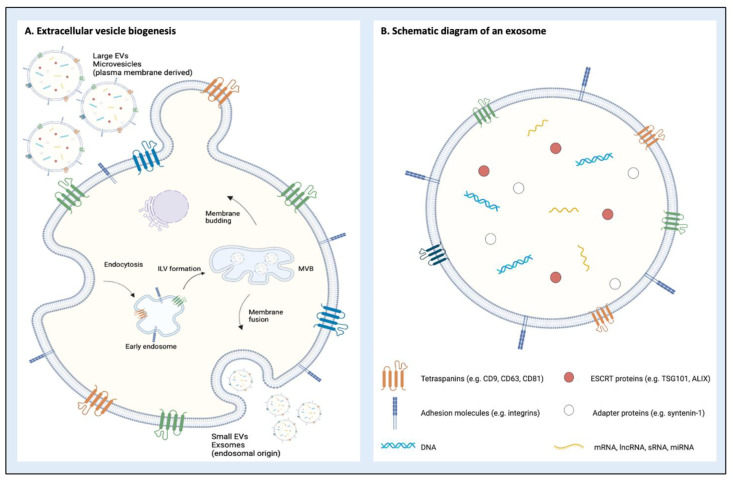
(**A**) Biogenesis of extracellular vesicles (EVs) demonstrating endosomal and membrane-budding pathways. Small EVs (exosomes) are generated through invagination of the cell membrane, which forms intraluminal vesicles (ILVs) that fuse with a multivesicular body (MVB). The MVB fuses with the cell membrane and the small EVs are exocytosed into the extracellular space. Large EVs (microvesicles) form through buddying of the cell membrane, which eventually “pinches” off to form vesicles. (**B**) A schematic of a small EV containing protein, DNA and RNA cargo with surface expression markers in the form of tetraspanins and integrins.

**Figure 2 pharmaceutics-15-00891-f002:**
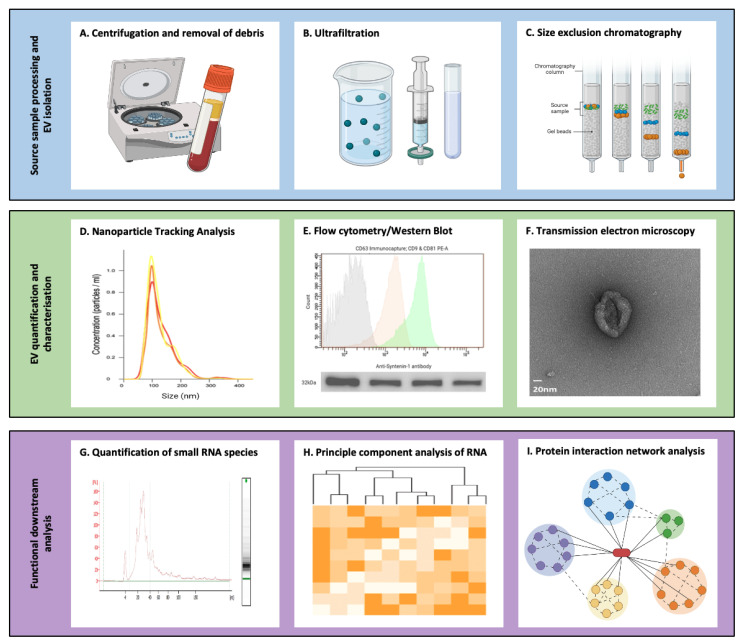
Methods of isolation and characterisation of extracellular vesicles (EVs). (**A**) One example of a source fluid is blood, for which centrifugation can be used to remove cellular debris and isolate serum. (**B**) Ultracentrifugation can be used to remove non-EV particles. (**C**) Size exclusion chromatography separates EVs from a sample based on their movement through a column of gel beads. (**D**) Nanoparticle tracking analysis quantifies the size and concentration of the particles isolated. (**E**) Flow cytometry and Western blotting can be used to confirm the presence of specific EV markers. (**F**) EVs can then be visualised and their morphology confirmed using transmission electron microscopy. (**G**) RNA can be isolated from EVs where smaller RNA species predominate. (**H**) Following next-generation sequencing, principal component analysis can be performed. (**I**) Proteomic analysis of EVs provides insight into the pathways they influence as signalling molecules.

**Figure 3 pharmaceutics-15-00891-f003:**
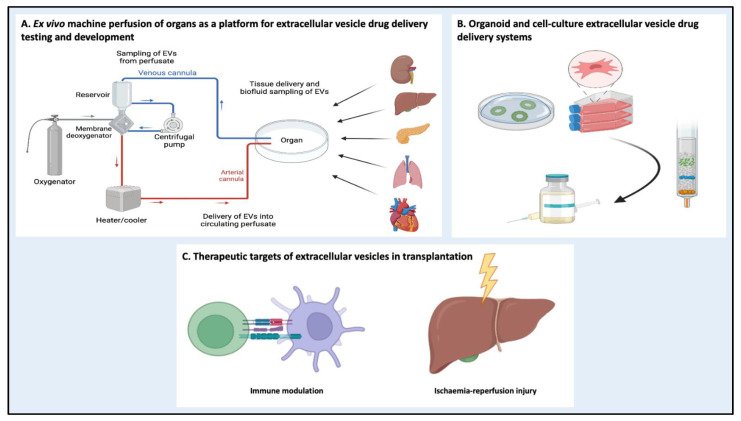
Organ transplantation represents a key area for extracellular vesicle (EV) drug delivery systems. (**A**) Ex vivo machine perfusion of organs provides a platform for EV delivery and allows sampling of organ biofluid post-treatment to monitor response. (**B**) Cell-culture and organoid systems are suitable EV production platforms. (**C**) Delivery of EVs during organ perfusion can modulate the immune system and target ischemia-reperfusion injury.

**Table 1 pharmaceutics-15-00891-t001:** **Classification of specific extracellular vesicle subtypes**. ALIX, ALG-2-interacting protein X; CD, cluster of differentiation; CX3CL1, C-X3-C motif chemokine ligand 1; CXCR4, chemokine receptor type; DNA, deoxyribonucleic acid; GAPDH, glyceraldehyde-3-phosphate dehydrogenase; lncRNA, long non-coding ribonucleic acid; MHC, mass histocompatibility complex; miRNA, micro-ribonucleic acid; mRNA, messenger ribonucleic acid; ROCK, Rho-associated protein kinase; TCR, T cell receptor; TSG101, tumour susceptibility gene 101.

Subtype	Origin	Size (nm)	Alternative Names	Composition	Biological Cargo
Exosomes	Endosome	50–150	Nanovesicles, proteosomes, exosome-like vesicles	*Membrane constituents:* Tetraspanins (e.g., CD9, CD63, CD81) Cell adhesion molecules (e.g., integrin) Cell-type-specific proteins (e.g., CXCR4, TCR, MHC-II) Lipids (e.g., phosphatidylserine, cholesterol) *Intravesicular components:* Signal transduction (e.g., Beta-catenin) Biogenesis factors (ALIX, TSG101, syntenin-1)	Enzymes (e.g., peroxidases), nucleic acids (e.g., miRNAs, mRNA, lncRNA)
Microvesicles	Plasma membrane	150–1000	Microparticles, oncosomes, shedding vesicles, blebbing vesicles	*Membrane constituents:* Tetraspanins (e.g., CD9, CD63, CD81) Cell adhesion molecules (e.g., integrin) Cell-type-specific proteins (e.g., CD14, MHC-II) Lipids (e.g., phosphatidylethanolamine) *Intravesicular components:* Signal transduction (e.g., ROCK) Cytoplasmic material (e.g., GAPDH) Cytoskeletal microtubules (e.g., actin, tubulin)	Nucleic acids (e.g., miRNAs, mRNA, lncRNA, DNA, histones)
Apoptotic bodies	Plasma membrane	500–2000	Apo-EVs	*Membrane constituents:* Lipids (namely phosphatidylserine) Cell adhesion molecules (e.g., CX3CL1) *Intravesicular components:* Signal transduction (e.g., ROCK)	Nucleic acids (including histones, large DNA fragments and some miRNAs), organelles

**Table 2 pharmaceutics-15-00891-t002:** Methods of extracellular vesicle isolation.

Methodology	Principle	Advantages	Disadvantages
Ultrafiltration	Biofluid is passed through a porous membrane to filter particles larger than a predetermined size	Time-effective; high yield	Pores blocked with contaminants; contamination (e.g., proteins and RNA); small particles left on pores; EVs damaged by force used
Differential ultracentrifugation	High centrifugal speeds are applied for sufficient time periods to allow individual EVs to travel to the bottom of the tube and accumulate as a pellet; however, the method is less efficient at pelleting smaller/less dense particles	Commonly used; replicable; convenient operation; no sample pre-treatment required	Time-consuming; requires larger volumes of biofluid; unpredictable co-isolation (e.g., lipoproteins); damage to and aggregation and loss of EVs
Density gradient ultracentrifugation	EVs are purified based on their buoyant density by using a medium such as iodixanol and centrifugation	Improved separation of EVs from protein complexes; replicable	Time-consuming; low yield; EV damage; co-isolation of non-EV particles of similar densities
Polymer precipitation	Volume-expanding polymers reduce the solubility of EVs in solution, with isolation following the subsequent low-speed centrifugation	High yield; time-efficient; commercial kits available	Unclear effects of polymers on downstream applications; coprecipitation of proteins with further protein removal kits needed
Size exclusion chromatography	EVs in solution loaded onto a gel bead column, with larger EVs passing around the gel beads and eluting from the column first, whilst smaller particles progress more slowly through the bead matrix and elute later	Vesicle structure and integrity preserved; high purity; reproducible	Time-consuming; post-isolation concentration steps required
Immunoaffinity	Immunocapture utilising beads conjugated with antibodies toward EV surface markers	High sensitivity and specificity; EV subtype separation possible	Expensive; low yield; elution techniques can affect EV integrity

**Table 3 pharmaceutics-15-00891-t003:** **Clinical trials utilising extracellular vesicles as therapeutics that are actively recruiting or completed**. ^a^ Trials currently listed on clinicaltrials.gov; trial names are the same as those found in the clinicaltrials.gov register. EV source is provided where possible, NA indicates that the source was not specified. ^b^ Where available, EV content has also been added. ^c^ Trial conclusions have been added where results are available. ANG1, angiopoietin 1; bFGF, basic fibroblast growth factor; CD24, cluster of differentiation marker 24; HGF, hepatocyte growth factor; IL-1B, interleukin-1-beta; NSCLC, non-small cell lung cancer; PDGF-AA, platelet-derived growth factor-AA; siRNA, small interfering ribonucleic acid; TGFb3, transforming growth factor beta-3; TIMP-1 and TIMP-2, tissue inhibitor of matrix metalloproteinases 1 and 2; VEGF, vascular endothelial growth factor; VEGFR, vascular endothelial growth factor receptor.

Number	Name ^a^	Condition	EV Source ^b^	Location	Phase	NCT Number ^c^
**Actively recruiting**						
1	Use of Autologous Plasma Rich in Platelets and Extracellular Vesicles in the Surgical Treatment of Chronic Middle Ear Infections	Chronic otitis media	Blood-derived (autologous)	Ljubljana, Slovenia	II/III	NCT04761562
2	Safety Evaluation of Intracoronary Infusion of Extracellular Vesicles in Patients With AMI	Myocardial infarction	Blood-derived	Minnesota, USA	I	NCT04327635
3	Autologous Serum-derived EV for Venous Trophic Lesions Not Responsive to Conventional Treatments (SER-VES-HEAL)	Venous ulcers	Blood-derived (autologous)	Turin, Italy	NA	NCT04652531
4	Bone Marrow Mesenchymal Stem Cell Derived EVs for COVID-19 Moderate-to-Severe Acute Respiratory Distress Syndrome (ARDS): A Phase III Clinical Trial	SARS-CoV-2	Bone marrow MSC-derived (cargo includes VEGFR, VEGF, ANG1, TIMP-1, TIMP-2, IL-1B, PDGF-AA, TGFb3, bFGF, HGF)	Texas, USA	III	NCT05354141
5	Safety and Efficacy of Injection of Human Placenta Mesenchymal Stem Cells Derived Exosomes for Treatment of Complex Anal Fistula	Fistula-in-ano	Human placenta MSC-derived	Tehran, Iran	I/II	NCT05402748
6	Allogenic Mesenchymal Stem Cell Derived Exosome in Patients With Acute Ischemic Stroke	Ischaemic stroke	Allogeneic MSC-derived (cargo enriched for miR-124)	Isfahan, Iran	I/II	NCT03384433
7	Efficacy and Safety of EXOSOME-MSC Therapy to Reduce Hyper-inflammation In Moderate COVID-19 Patients (EXOMSC-COV19)	SARS-CoV-2	MSC-derived	Indonesia	II/III	NCT05216562
8	A Clinical Study of Mesenchymal Progenitor Cell Exosomes Nebulizer for the Treatment of Pulmonary Infection	Pulmonary infection	Mesenchymal progenitor MSC-derived	Shanghai, China	I/II	NCT04544215
9	Study Investigating the Ability of Plant Exosomes to Deliver Curcumin to Normal and Colon Cancer Tissue	Colon cancer	Plants (cargo of curcumin)	Kentucky, USA	I	NCT01294072
10	Evaluation of the Safety of CD24-Exosomes in Patients With COVID-19 Infection	SARS-CoV-2	CD24-expressing 293-TREx™ cells (EVs enriched for CD24)	Tel Aviv, Israel	I	NCT04747574
11	Clinical Efficacy of Exosome in Degenerative Meniscal Injury (KNEEXO)	Degenerative meniscal injury	MSC-derived	Eskisehir, Turkey	II	NCT05261360
12	The Effect of Stem Cells and Stem Cell Exosomes on Visual Functions in Patients With Retinitis Pigmentosa	Retinitis pigmentosa	Wharton jelly- derived mesenchymal stem cells	Kayseri, Turkey	II/III	NCT05413148
14	Effect of UMSCs Derived Exosomes on Dry Eye in Patients With cGVHD	Dry eye	Umbilical MSC- derived	Guangdong, China	I/II	NCT04213248
15	iExosomes in Treating Participants With Metastatic Pancreas Cancer With KrasG12D Mutation	Pancreatic cancer	MSC-derived (cargo of siRNA against KrasG12D)	Texas, USA	I	NCT03608631
16	Safety and Efficacy of Exosomes Overexpressing CD24 in Two Doses for Patients With Moderate or Severe COVID-19	SARS-CoV-2	CD24-expressing 293-TREx™ cells (EVs enriched for CD24)	Athens, Greece	II	NCT04902183
17	Safety and Effectiveness of Placental Derived Exosomes and Umbilical Cord Mesenchymal Stem Cells in Moderate to Severe Acute Respiratory Distress Syndrome (ARDS) Associated With the Novel Corona Virus Infection (COVID-19)	SARS-CoV-2	Umbilical cord MSC-derived (cargo of unspecified growth factors)	Missouri, USA	I	NCT05387278
18	An Open, Dose-escalation Clinical Study of Chimeric Exosomal Tumor Vaccines for Recurrent or Metastatic Bladder Cancer	Bladder cancer	Chimeric exosomal tumour vaccine	Shanghai, China	I	NCT05559177
19	A Study of exoASO-STAT6 (CDK-004) in Patients With Advanced Hepatocellular Carcinoma (HCC) and Patients With Liver Metastases From Primary Gastric Cancer and Colorectal Cancer (CRC)	Hepatocellular carcinoma, metastatic gastric and colorectal cancer	Bioengineered (cargo of STAT6 anti-sense oligonucleotide)	California, USA	I	NCT05375604
**Completed**						
1	Efficacy of Platelet- and Extracellular Vesicle-rich Plasma in Chronic Postsurgical Temporal Bone Inflammations (PvRP-ear)	Chronic inflammation of temporal bone post-surgery	Blood-derived (autologous)	Ljubljana, Slovenia	NA	NCT04281901
2	Extracellular Vesicle Infusion Treatment for COVID-19 Associated ARDS (EXIT-COVID19)	SARS-CoV-2	Bone marrow MSC-derived	Alabama, USA	II	NCT04493242
3	Safety and Tolerability Study of MSC Exosome Ointment	Psoriasis	MSC-derived (cargo of VEGFR, VEGF, ANG1, TIMP-1, TIMP-2, IL-1B, PDGF-AA, TGFb3, bFGF, HGF)	Singapore	I	NCT05523011
4	A Pilot Clinical Study on Inhalation of Mesenchymal Stem Cells Exosomes Treating Severe Novel Coronavirus Pneumonia	SARS-CoV-2	MSC-derived	Wuhan, China	I	NCT04276987 (conclusion: inhalation of EVs up to a total amount of 2.0 × 10^9^ was feasible and functioned well, with no evidence of prespecified adverse events, immediate clinical instability or dose-relevant toxicity at any of the doses tested. This safety profile was seemingly followed by CT imaging improvement within 7 days)
5	Intra-discal Injection of Platelet-rich Plasma (PRP) Enriched With Exosomes in Chronic Low Back Pain	Chronic lower back pain	Blood derived (autologous)	Uttarakhand, India	I	NCT04849429
6	Evaluation of Safety and Efficiency of Method of Exosome Inhalation in SARS-CoV-2 Associated Pneumonia (COVID-19EXO)	SARS-CoV-2	MSC-derived	Volga, Russia	I/II	NCT04491240
7	A Tolerance Clinical Study on Aerosol Inhalation of Mesenchymal Stem Cells Exosomes In Healthy Volunteers	Nil	Adipose MSC- derived	Shanghai, China	I	NCT04313647 (conclusion: all volunteers tolerated EV nebulization well, with no serious adverse events observed. The authors suggested that nebulised EVs could be a promising therapeutic strategy in lung injury diseases)
8	Plant Exosomes ± Curcumin to Abrogate Symptoms of Inflammatory Bowel Disease	Inflammatory bowel disease	Plants (cargo of curcumin)	Kentucky, USA	NA	NCT04879810
9	Edible Plant Exosome Ability to Prevent Oral Mucositis Associated With Chemoradiation Treatment of Head and Neck Cancer	Oral mucositis	Grapes	Kentucky, USA	I	NCT01668849
10	Trial of a Vaccination With Tumor Antigen-loaded Dendritic Cell-derived Exosomes (CSET 1437)	Non-small cell lung cancer	Dendritic cell derived (cargo of melanoma- associated antigen)	Villejuif, France	II	NCT01159288 (conclusion: EVs boost the natural killer cell arm of antitumour immunity in patients with advanced NSCLC)

## Data Availability

No new data were created or analysed in this study. Data sharing is not applicable to this article.

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
