# Peer review of "Extracellular Vesicles as Drug Delivery Systems in Organ Transplantation: The Next Frontier"

_pharmaceutics, 2023, doi:10.3390/pharmaceutics15030891_

Round 1

Reviewer 1 Report

In the manuscript by HVM et al, “Extracellular vesicles as drug delivery systems in organ transplantation: the next frontier”. In this manuscript, authors have tried to cover the work related to the EVs for organ transplantation, however, the following comments/suggestions must be addressed before publication.

-Major comment: Manuscript is poorly formatted, with several different font types and font sizes which is distracting during reading.

-Authors need to classify different types of EVs with more distinct identification, maybe a small table will help and with main biological cargos.

-Isolation of EVs can be simply put in one table, as it is not much important which is easily available in the literature.

(Ref. https://doi.org/10.1016/bs.acc.2020.08.006)

-Authors should more focus on EVs as therapeutics for organ transplantation with more research study support, which is lacking

-From literature support, authors need to explain in more detailed about biological cargos responsible for therapeutic activity. This information can be explained in tabular form.

Author Response

Reviewer 1

In the manuscript by HVM et al, “Extracellular vesicles as drug delivery systems in organ transplantation: the next frontier”. In this manuscript, authors have tried to cover the work related to the EVs for organ transplantation, however, the following comments/suggestions must be addressed before publication.

-Major comment: Manuscript is poorly formatted, with several different font types and font sizes which is distracting during reading.

Thank you for the comment and we apologise for any formatting issues. This has been corrected and the font type made uniform. The differences in font size are to subset headings which have been kept in for clarity.

-Authors need to classify different types of EVs with more distinct identification, maybe a small table will help and with main biological cargos.

Thank you for the comment and we agree that a table on EV subtypes and their cargo would be beneficial to the manuscript. This has been added as Table 1. We have also added the sentence: ‘consensus on specific EV subtype markers remains elusive, therefore, assigning an EV to a specific biogenesis pathway is challenging’. This is the justification for subsequent adoption of the International Society of Extracellular Vesicles guidelines on reporting of EVs, which suggest using the terminology ‘small EVs’ and ‘medium/large EVs’ (reference 18).

-Isolation of EVs can be simply put in one table, as it is not much important which is easily available in the literature.

(Ref. https://doi.org/10.1016/bs.acc.2020.08.006)

Thank you for the comment and for providing the example reference highlighting the ease of access to literature concerning methods of EV isolation. We have removed the ‘Isolation of EVs’ subheading and added in a sentence to explain the thought processes behind choice of isolation as follows: “As there is no single optimal method of isolation, the choice of technique is based on the downstream application and scientific question being addressed.” We have also provided a table of isolation methods as requested and provided a reference to this in the text as Table 2.

-Authors should more focus on EVs as therapeutics for organ transplantation with more research study support, which is lacking.

Thank you for your comment, please see our response to the comment below which provides an amendment achieving both changes.

-From literature support, authors need to explain in more detailed about biological cargos responsible for therapeutic activity. This information can be explained in tabular form.

Thank you for the comment and suggestion. We have included a table of EV subtypes with their biological cargos, as well increasing the discussion surrounding EVs as therapeutics in organ transplantation. This includes description of the biological processes targeted and the mechanisms by which EV cargo might mediate its effect, therefore, we feel a further table (in addition to the current three) would not be of great benefit here. Additionally, one important conclusion is that further exploration of biologically active EV cargo is needed, as well as the timing of EV-mediated miRNA delivery in the ischaemia-reperfusion-injury process, which has been highlighted in the conclusion to that section.

Reviewer 2 Report

The authors provided a well-documented overview of Ev's role and application in in organ transplantation. In addition to this, the review could represent a really interesting point of view in a field so dynamic and rich in potential future applications of exosomes as nanotherapeutics. If the article is well written, the introduction section could be improved by adding some recent works could be improved with a more general point of view about the application of EVs research in other fields of research, adding some recent works related to the importance of exosomes in other diseases (PMID: 34839044). A point that should be fixed is related to Figure 3: the text of cartoon should be increased in size. The conclusions could be improved and enriched by a discussion related to the need for new technologies for the association of a specific marker with an exosome subtype and the exosome subtype to a particular function and/or group of functions (PMID: 35141731 and others).

Author Response

The authors provided a well-documented overview of Ev's role and application in in organ transplantation. In addition to this, the review could represent a really interesting point of view in a field so dynamic and rich in potential future applications of exosomes as nanotherapeutics.

Thank you to the reviewer for such a positive response and for recognising the exciting potential of EVs in transplantation.

If the article is well written, the introduction section could be improved by adding some recent works could be improved with a more general point of view about the application of EVs research in other fields of research, adding some recent works related to the importance of exosomes in other diseases (PMID: 34839044).

Thank you for the comment and we agree the highlighting of EVs in other disease processes is important. These have been added along with the reference recommended.

A point that should be fixed is related to Figure 3: the text of cartoon should be increased in size.

Thank you for the comment, this has been amended.

The conclusions could be improved and enriched by a discussion related to the need for new technologies for the association of a specific marker with an exosome subtype and the exosome subtype to a particular function and/or group of functions (PMID: 35141731 and others).

Thank you for the comment. We agree and have included a discussion related to current attempts to isolate and characterise exosomes, along with reinforcing the need for new technologies to link subpopulation with biological function.

Round 2

Reviewer 1 Report

Authors have addressed the comments and suggestions, so recommended to accept it.

Thanks!

Author Response

Thank you to the reviewer for their acceptance of our revisions and positive response to our submission.